# Epidemiology, literacy, risk factors, and clinical status of oral cancer in East Africa: A scoping review

Kafayat Aminu[1]*, Timothy Olukunle Aladelusi[2,3], Akinyele Olumuyiwa Adisa[4,5], Chiamaka Norah Ezeagu[6], Afeez Abolarinwa Salami[2,6,7,8], Jacob Njideka Nwafor[7,8,9,10], Peace Uwambaye[11], Jimoh Amzat[12,13,14], Julienne Murererehe[11]*, Semeeh Akinwale Omoleke[15], Mohammed Abdulaziz[16], Ruwan Duminda Jayasinghe[6,12], Kehinde Kazeem Kanmodi[6,7,11,12]*

1 Centre for Child and Adolescent Mental Health, University College Hospital, Ibadan, Nigeria, 2 Department of Oral and Maxillofacial Surgery, University College Hospital, Ibadan, Nigeria, 3 Department of Oral and Maxillofacial Surgery, University of Ibadan, Ibadan, Nigeria, 4 Department of Oral Pathology and Oral Medicine, University College Hospital, Ibadan, Nigeria, 5 Department of Oral Pathology and Oral Medicine, University of Ibadan, Ibadan, Nigeria, 6 Faculty of Dentistry, University of Puthisastra, Phnom Penh, Cambodia, 7 Campaign for Head and Neck Cancer Education (CHANCE) Programme, Cephas Health Research Initiative Inc., Ibadan, Nigeria, 8 Department of Public Health Dentistry, Manipal Academy of Higher Education, Manipal, Karnataka, India, 9 Faculty of Medicine, University of Puthisastra, Phnom Penh, Cambodia, 10 Division of Medicine, University of Nottingham NHS Hospitals Foundation Trust, Nottingham, United Kingdom, 11 Department of Preventive and Community Dentistry, University of Rwanda, Kigali, Rwanda, 12 School of Health and Life Sciences, Teesside University, Middlesbrough, United Kingdom, 13 Department of Sociology, Usmanu Danfodiyo University, Sokoto, Nigeria, 14 Department of Sociology, University of Johannesburg, Johannesburg, South Africa, 15 Department of International Public Health, EUCLID University, Bangui, Central Africa Republic, 16 Division of Disease Control and Prevention, Africa Centres for Disease Control and Prevention, Adis Ababa, Ethiopia

* j.murererehe@ur.ac.rw (JM); bolkaf@yahoo.com (KA); kanmodikehinde@yahoo.com (KKK)

## Abstract

### Background

Oral cancer (OC) is a topical public health issue in East Africa due to increasing incidence of the disease. Public health efforts to address the oral cancer burden depends largely on the available empirical evidence. Hence, this scoping review aims to map the existing empirical evidence on oral cancer in East African countries.

### Methods

The Preferred Reporting Items for Systematic Review and Meta-analysis Extension for Scoping Reviews (PRISMA-ScR) was used as a guideline for reporting this scoping review. Additionally, we ensured quality assessment of the methodology and reporting process of this study using the AMSTAR 2 checklist. We conducted a systematic search of nine research databases on 17th November 2023, and reviewed studies published in English from year 2000 to 17th November 2023. The team developed data extraction form and data extraction was done by two reviewers. Thematic analyses were conducted manually and presented in texts, tables and flow chart.

**Data availability statement:** All relevant data are within the paper and its Supporting Information files.

**Funding:** The author(s) received no specific funding for this work.

**Competing interests:** The authors have declared that no competing interests exist.

**Abbreviations:** BL, Burkitt's Lymphoma; Fig, Figure; GPs, General Practitioners; IARC, International Agency for Research on Cancer; KS, Kaposi's sarcoma; NHL, non-Hodgkin's lymphoma; NOR, Nucleolar Organizer Region; NPC, Nasopharyngeal Carcinoma; OC, Oral Cancer; OSCC, oral squamous cell carcinoma; OSCC, oral squamous cell carcinoma; OS, Osteosarcoma; PRISMA-ScR, Preferred Reporting Items for Systematic Review and Meta-analysis Extension for Scoping Reviews; RMS, Rhabdomyosarcoma; SCC, Squamous Cell Carcinoma

## Results

Only 30 full manuscripts were included in this review. Twenty-nine out of 30 studies were either hospital- or clinic-based while two were community-based. Only four studies showed gaps and obvious disparities in awareness and knowledge levels across East Africa, however, higher levels of awareness were reported among dentists and dental patients relative to the general population. Most neoplasms were presented and diagnosed late. The review finding also highlighted the significant impact of Toombak use on the oral microbiome composition, potentially contributing to oral cancer risks. Further, this review elucidated the prognostic relevance of PD-L1 expression at the invasive tumor front and microbial composition, with *Candida* correlating with adverse prognosis and *Malassezia* showing associations with improved survival rates. Also, Toombak usage, tumor staging, and mucosal field alterations emerged as predictors of local recurrence, while lymph node involvement and extranodal extension were associated with regional recurrence among Sudanese cohorts. Finally, a few studies undertook an evaluation of instrument validity for OC detection, revealing promising outcomes concerning diagnostic accuracy and instrument reliability.

## Conclusions

There is a dire need for targeted interventions and early detection strategies tailored to the unique epidemiological and clinical profiles of oral and maxillofacial tumors in East Africa. Public health interventions aimed at curbing the prevalence of Toombak use and promoting healthier lifestyle choices to reduce the oral diseases incidence in Sudan and other regions where these behaviors are prevalent remain germane.

## Introduction

Oral cancer (OC) develops in the tissues of the mouth, lips, and oropharynx [1,2]. Oral cancer has become a significant public health issue in East Africa due to increasing incidence of the disease with the most commonly diagnosed being squamous cell carcinoma, involving the lips and tongue [3]. It is predictable that over 7,000 new cases of oral cancer are identified in the region each year, with a mortality rate of 70% [4]. The rising incidence of oral cancer in the region is largely due to increased exposure to risk factors, including tobacco and alcohol use, poor oral hygiene, and viral infections such as human papillomavirus (HPV) [3]. According to a report by the International Agency for Research on Cancer (IARC), East Africa has one of the highest incidences of oral cancer in the Africa [4]. In Tanzania, for instance, oral cancer accounts for approximately 1.2% of all cancer cases. According to the latest WHO data published in 2020, oral cancer deaths in Kenya reached 1.8% of total cancer-related deaths.

Late detection of oral cancer is a common occurrence in East Africa and a challenge, and this is largely due to limited access to cancer screening and treatment facilities [3,5]. Most cases of oral cancer in the region are, therefore, detected at advanced stages, when the cancer is already locally advanced and might have already spread to other parts of the body, making it difficult to treat, resulting in high mortality rates [6].

There is increasing efforts to address the oral cancer burden in East Africa and such initiatives are focused on prevention, early detection, treatment, and rehabilitation [7]. These efforts include public education campaigns to raise awareness about the risks of tobacco and alcohol use, as well as the importance of maintaining good oral hygiene. Screening programs

are also being employed to recognise cases of oral cancer at an early stage when intervention is more likely to be effective [8,9].

Nonetheless, considerable limitations remain in the fight against oral cancer in East Africa, including limited research, constrained resources and insufficient healthcare infrastructure [3,5]. Dealing with these limitations will require persistent efforts from governments, healthcare providers, and other stakeholders, including researchers, to enhance prevention, diagnosis, and treatment of oral cancer in the region. As a part of the initial steps in the fight towards oral cavity cancer control or reduction in East Africa, there is need to map the existing empirical evidence concerning the disease in the sub-region.

This study aims to conduct a scoping review to describe, synthesize, and appraise the existing empirical evidence on the epidemiology, literacy, risk factors, and clinical status of oral cancer in East African countries which includes Burundi, Comoros, Djibouti, Ethiopia, Eritrea, Kenya, Rwanda, Seychelles, Somalia, South Sudan, Sudan, Tanzania, and Uganda, according to the African Development Bank's classification [10]. The findings obtained from this review will provide insights which will set the pace for future oral cancer-related research and policies in the sub-region.

## Methods

### Study design

A scoping review design was adopted for this study as it aims to describe, synthesize, and appraise all the available empirical evidence on oral cancer in East Africa [11]. The Arksey and O'Malley's framework for conducting scoping reviews was adopted in this review, thus this review's methodology had these steps: 1) Identification of the research question; 2) Identification of relevant studies; 3) Selection of relevant studies for inclusion; 4) Charting of data; and 5) Collation, summary and reporting of results [12].

The Preferred Reporting Items for Systematic Review and Meta-analysis Extension for Scoping Reviews (PRISMA-ScR) was used as a guideline for reporting this scoping review [13]. Additionally, to ensure quality assessment of the methodology and reporting process of this study, the guidelines provided by the AMSTAR 2 checklist were also utilised [14].

### Identification of the research question

The research question for this scoping review was developed using the PCC (P – Population; C – Concept; and C – Context) framework [15], here the population of interest was human population, the concepts of interest were epidemiology, literacy, risk factors, and clinical status (including diagnosis, clinical features, and treatment outcomes), and the context of interest was East Africa. Based on this framework, this review's research question was: "What is the nature and extent of empirical findings available in existing literature on Oral cancer in East African countries?"

### Identification of relevant studies

To identify relevant studies for this scoping review, we conducted a systematic search of 9 research databases, including PubMed, SCOPUS, AMED – The Allied and Complementary Medicine Database, APA PsycArticles, APA PsycInfo, CINAHL Ultimate, Dentistry & Oral Sciences Source, Psychology and Behavioral Sciences Collection, and SPORTDiscus with Full Text. (S1 to S3 Tables; Supplementary file). The search conducted on 17 November 2023, and it was limited to studies published in English. The keywords used for the search were obtained from a search of the Medical Subject Heading (MeSH) dictionary together with a review of

multiple structured literature reviews on oral cancer/East Africa [16–19]. These keywords included "oral cancer", "oral squamous cell carcinoma", "oropharyngeal cancer", "oral cavity cancer", "mouth cancer", "oral cavity cancer", "oral malignant neoplasia", "East Africa", "Burundi", "Comoros", "Djibouti", "Ethiopia", "Eritrea", "Kenya", "Rwanda", "Seychelles", "Somalia", "South Sudan", "Sudan", "Tanzania", and "Uganda". Truncation ("*") and Boolean operators ("OR" and "AND") were used to combine the search terms during the search. The search was specifically targeted at the titles, abstracts, and keywords of the literature to ensure that only relevant studies were identified. S1 to S3 Tables (Supplementary file) shows the search strings used to search each of the databases.

## Selection of relevant studies for inclusion

The Rayyan web application was used to remove duplicate copies from the retrieved literature. The de-duplicated copies were then screened to ensure eligibility for inclusion into the study.

**Eligibility.** Only those literatures that met the following criteria were included into this scoping review:

- Original research articles published in peer-reviewed journals.

- Articles reporting empirical findings on oral cancer in East Africa countries (this includes Burundi, Comoros, Djibouti, Ethiopia, Eritrea, Kenya, Rwanda, Seychelles, Somalia, South Sudan, Sudan, Tanzania, and/or Uganda).

- Articles published in English.

- Articles published between 2000 and 17 November 2023 (the literature search date).

- Articles with accessible full text.

Those literatures that did not meet all the above inclusion criteria were excluded from this scoping review.

The screening process involved two stages. Each stage of the screening process was carried out by two independent reviewers and based on strict adherence to the review's inclusion criteria. In the first stage, titles and abstracts were screened to identify those literatures that met the review's inclusion criteria at face value. Those literatures identified from the first stage of the screening process were then moved to the second stage for full text screening. Any existing discrepancies in either of the two screening stages were resolved by the entire team through discussion, and no conflict was unresolved.

## Quality appraisal

The appraisal of the quality of the included articles was done by two independent reviewers, using the Mixed Methods Appraisal Tool 2018 version. This appraisal tool uses seven questions to appraise articles and the composition of these questions depend on the study design (qualitative study design, quantitative nonrandomised study design, quantitative randomised study design, or mixed methods study design) of the article under appraisal. There are three possible responses to each question of the appraisal tool: and "No", "I can't tell", and "Yes". In the appraisal process, a response of "No" was scored zero, a response of "I can't tell" was scored 0.5, and a response of "Yes" was scored 1. Since the appraisal tool has seven questions in total for each study design, each appraised article was graded on a scale of 0 to 7. Any appraised literature with a cumulative score of < 3.5/7 was graded as "below average" quality, anyone with a cumulative score of 3.5/7 was graded as "average" quality, and anyone with a cumulative score of > 3.5/7 was graded as "above average" quality.

### Charting of data

A data extraction form was collaboratively developed by the team and used to chart data from each of the included literature. Data from the selected literature were organized into distinct categories, including author and year of publication, country of study, study aim, study design, study setting, study population, study instrument, relevant findings, and limitations. The data extraction was collectively done by two reviewers.

### Collation, summary and reporting of results

Numerical and thematic analyses of the included literature were conducted manually, and presented in texts, tables, and flow chart. In the thematic analysis, codes were generated by familiarisation and critical examination of how texts address the research question [19]. The generated codes were grouped, reviewed and discussed by the entire team, leading to the development of categories and themes [20,21].

## Results

A total of 177 literatures (PubMed = 63, SCOPUS = 114, AMED – The Allied and Complementary Medicine Database = 0, APA PsycArticles = 0, APA PsycInfo = 0, CINAHL Ultimate = 0, Dentistry & Oral Sciences Source = 0, Psychology and Behavioral Sciences Collection = 0, SPORTDiscus with Full Text = 0) were retrieved from the database search. Fifty-six published literature were duplicate records and were deleted. After screening of the remaining 121 literature, only 30 [5,6,22–49] full length original research articles were finally included into the review (Fig 1).

### Quality appraisal outcome

The outcomes of the appraised 30 articles, using the Mixed Methods Appraisal Tool are depicted in Fig 2 (below) and S5 to S8 Tables (supplementary file). More than half (56.7%; 17/30) of the appraised articles had an above average quality, six (20%) had average quality, and seven (23.3%) had below average quality.

### Distribution of the included articles by year of publication

None of the articles included in the review were published prior to the year 2000 as shown in Fig 3. The number of publications on OC in East Africa increased over time (though slight decline was noticeable), with 5 articles from 2000–2004, 4 from 2005–2009, 7 from 2010–2014, 6 from 2015–2019, and peaking at 8 from 2020–2023. The shortest period with the highest number of publications was from 2021 to 2023, with eight articles published during this time. This trend suggests a growing interest and rising research activity in the topic, resulting in the highest output in the most recent years

### Distribution of the included articles by country of study

Sudan stands out significantly with the highest number of entries. About two-third (19 of 30) of the reviewed articles was conducted in Sudan [22–26,28,29,32–37,41,43,44,47–49], four reported samples from Kenya [5,30,31,46], two were done in Uganda [27,45] and one in Rwanda [6]. Others are multi-national studies involving countries within and outside East Africa. One of these was done in Sudan and Sri Lanka [40], another one used sample from Sudan, Scandinavia, USA, and UK [38] while the remaining 2 were conducted in Sudan and Norway [39,42] (Fig 4).

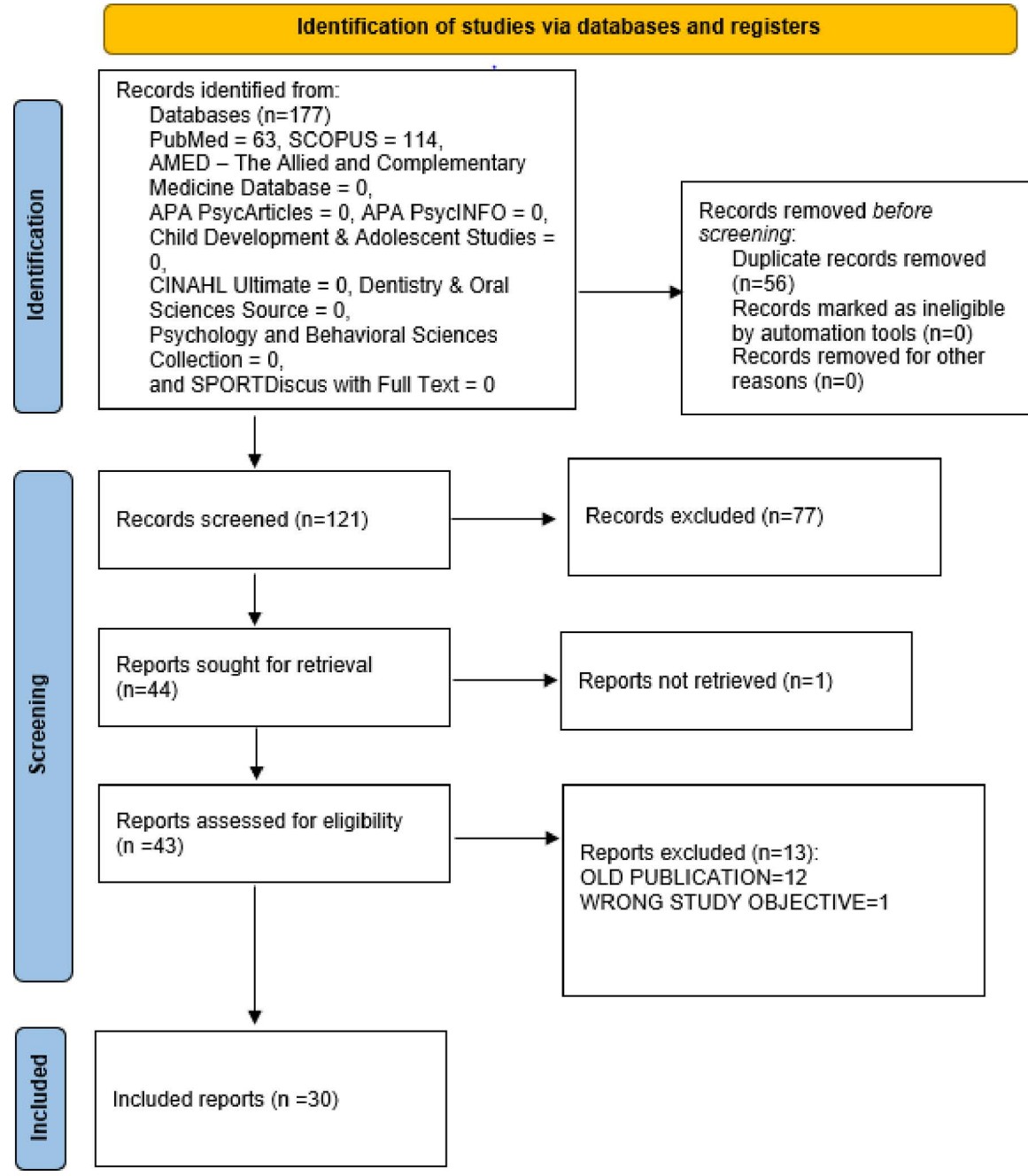

**Fig 1. PRISMA 2020 flowchart of included literature.**

## Distribution of the included articles by study design

The reviewed studies encompassed various designs. One was general cohort study, three were case-control, four were cross-sectional descriptive study, six were retrospective cohort design, eight studies each adopted prospective cohort and cross-sectional analytical design (Fig 5).

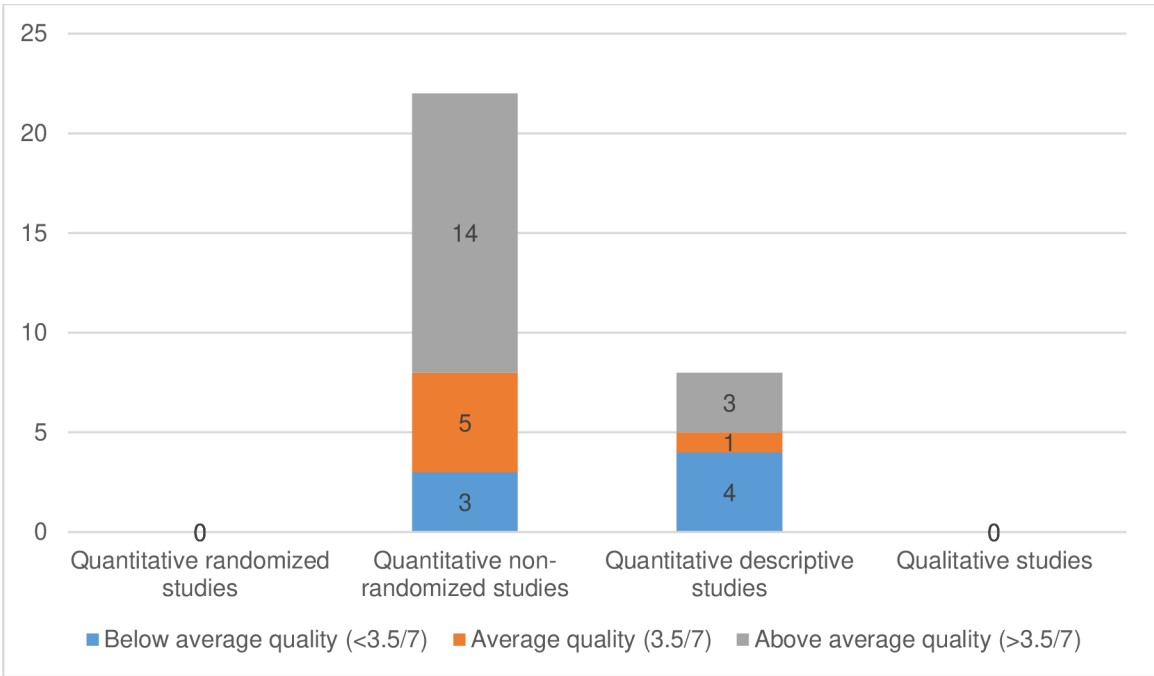

**Fig 2. Outcomes of the quality appraisals of the included articles using the Mixed Methods Appraisal Tool.**

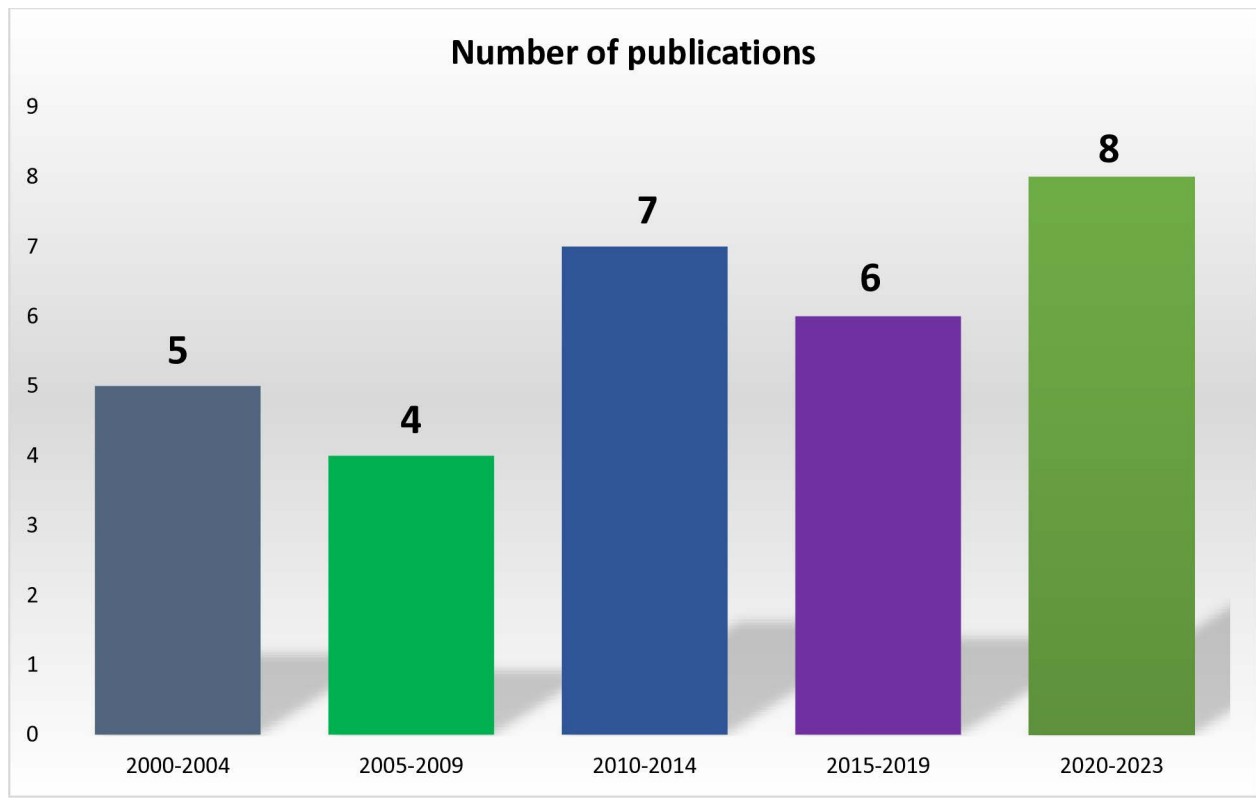

**Fig 3. Distribution of the included studies by year of publication.**

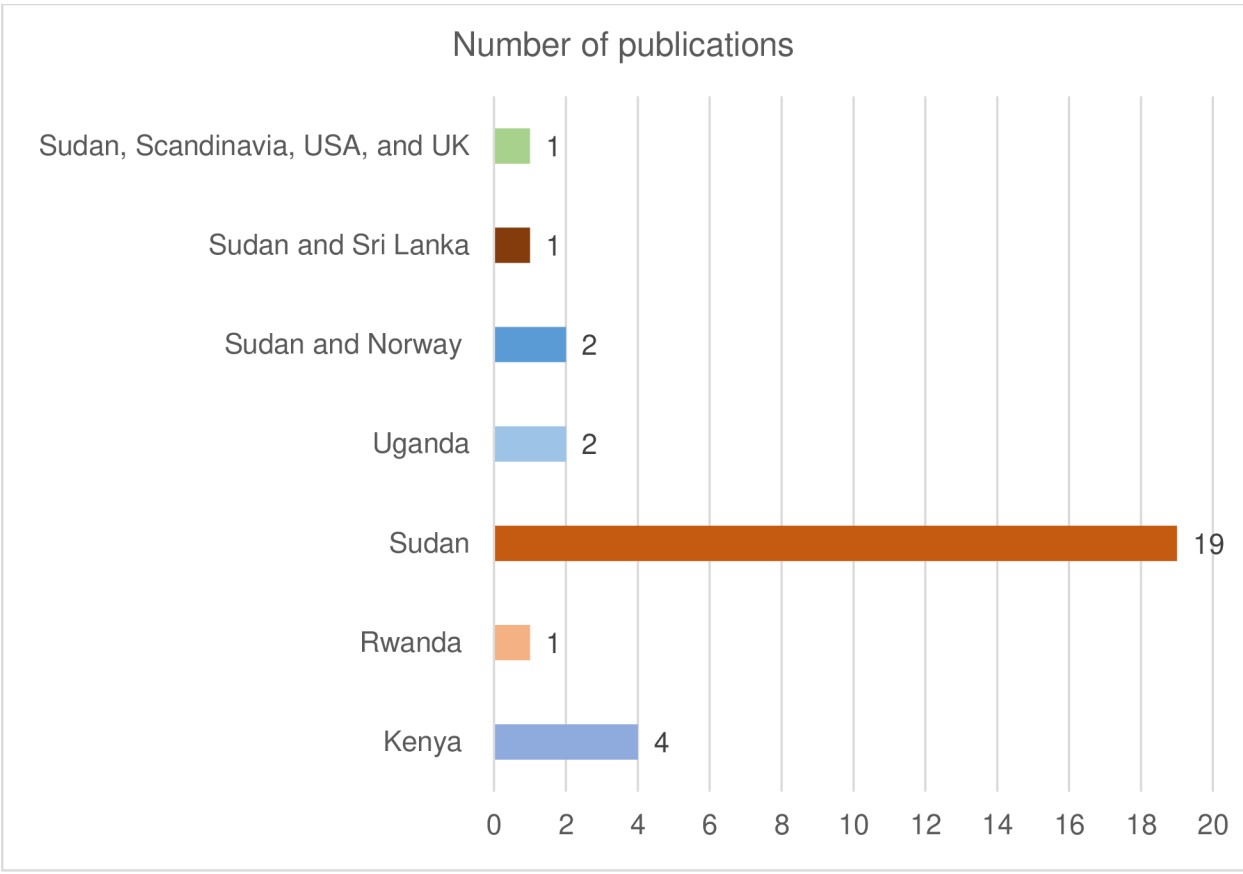

**Fig 4. Distribution of included studies by country of publication.**

### Distribution of the included articles by study setting

Twenty-eight of the 30 articles selected were clinic- or hospital-based, only two were community-based.

### Sample distribution of the included articles

Sample size in the included articles ranged between 22 and 1018. In total, 6157 persons were sampled, including 3304 male and 2510 female. However, three studies [26,38,42] did not specify male-to-female ratio in their samples.

### Distribution of included articles by research objectives

The majority of the articles included in this review focused on molecular and cellular mechanisms of OC (n−11, 37%). This was followed by those on clinical features and manifestations of OC (n−4, 13%) awareness and knowledge of OC (n−4, 13%), and epidemiological trends (n−4, 13%). There is less emphasis on OC risk factors (n−1, 3%) (Fig 6).

### Empirical findings

The empirical findings obtained from the reviewed articles were grouped under the following thematic areas: epidemiology of OC; literacy on OC; risk factors of OC; diagnosis of OC; and clinical status of diagnosed OC cases in East Africa (Table 1).

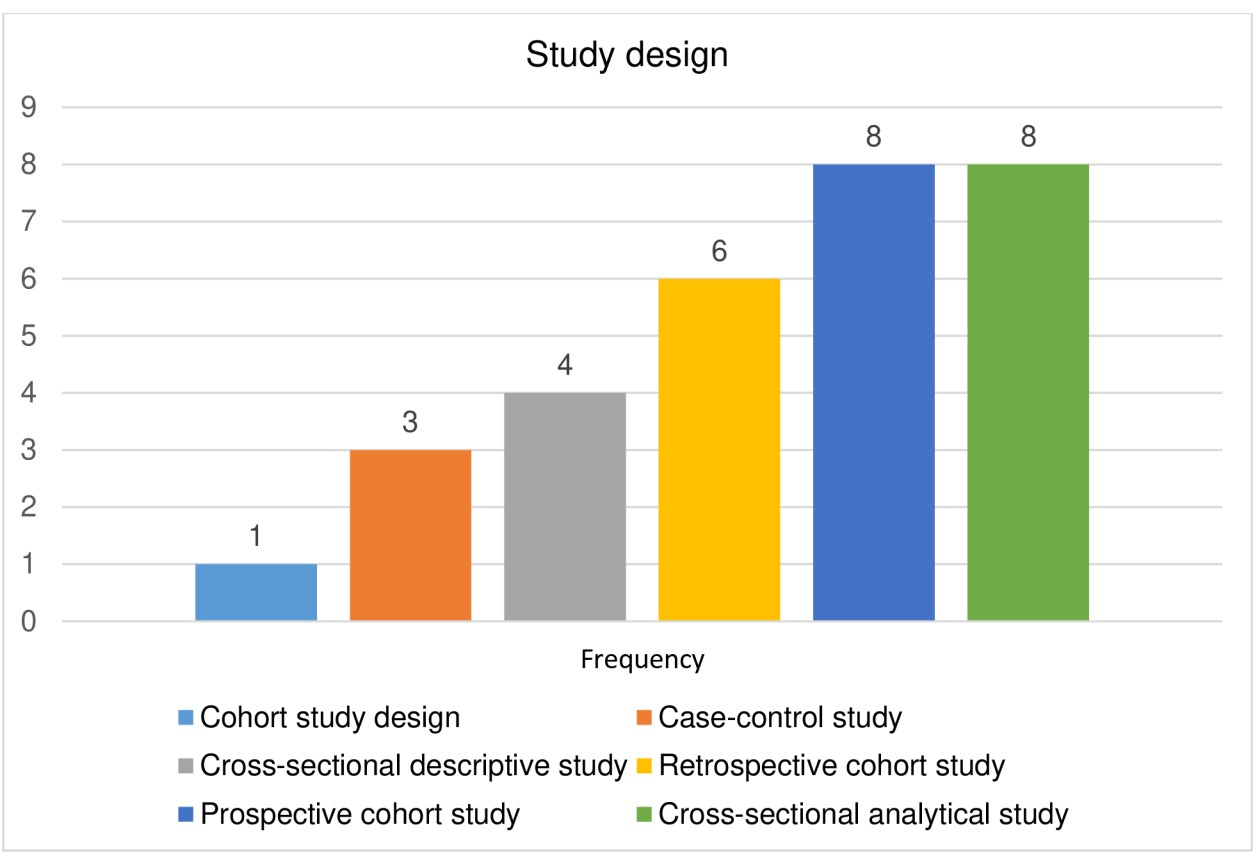

**Fig 5. Distribution of included articles by study design.**

**Epidemiology of oral cancer in East Africa.** Only few of the reviewed studies reported the prevalence of OC among East African countries. To start with, the all-time prevalence (3.6%) and annual frequency (1.5–7%) of OC in Kenya were 3.6% and 1.5–7%, respectively [5]. In another study among Kenyans, many patients (53.83%) tested positive for oral and maxillofacial tumours of which majority (60.51%) were malignancies [31].

In Sudan, the most common malignant tumour reported by the studies include locoregional lymph node metastases (69.3%) [44]. However, oral squamous cell carcinoma (OSCC) was the most diagnosed in both Sudan [23,32,47] and Kenya [31].

Prevalence of OSCC in Sudan ranged from 45.1% to 73% [23,32,47] while 59.5% prevalence was reported in Kenya [31].

Age and gender differences were also observed in the occurrence of OC in East Africa. Orofacial (benign and malignant) tumours were more prevalent in Kenyan men than women [5]. Age also played a major factor in OC diagnosis as most cases (89.7%) were recorded in patients 40 years and older. Also in Kenya, gender influenced the site of infection as men were more affected in all sites, except the lips which was the common site of infection among females. Similarly, more females were diagnosed with KS (54%), and OSCC (67%) whereas more male had NHL (86%) in Kenya [5].

In Sudan, OC incidence was found to increase with age OC predominantly affected males (62%) compared to females (38%) [28]. Incidence of benign tumours was similar across genders, while malignant lesions were more common in male Sudanese [28].

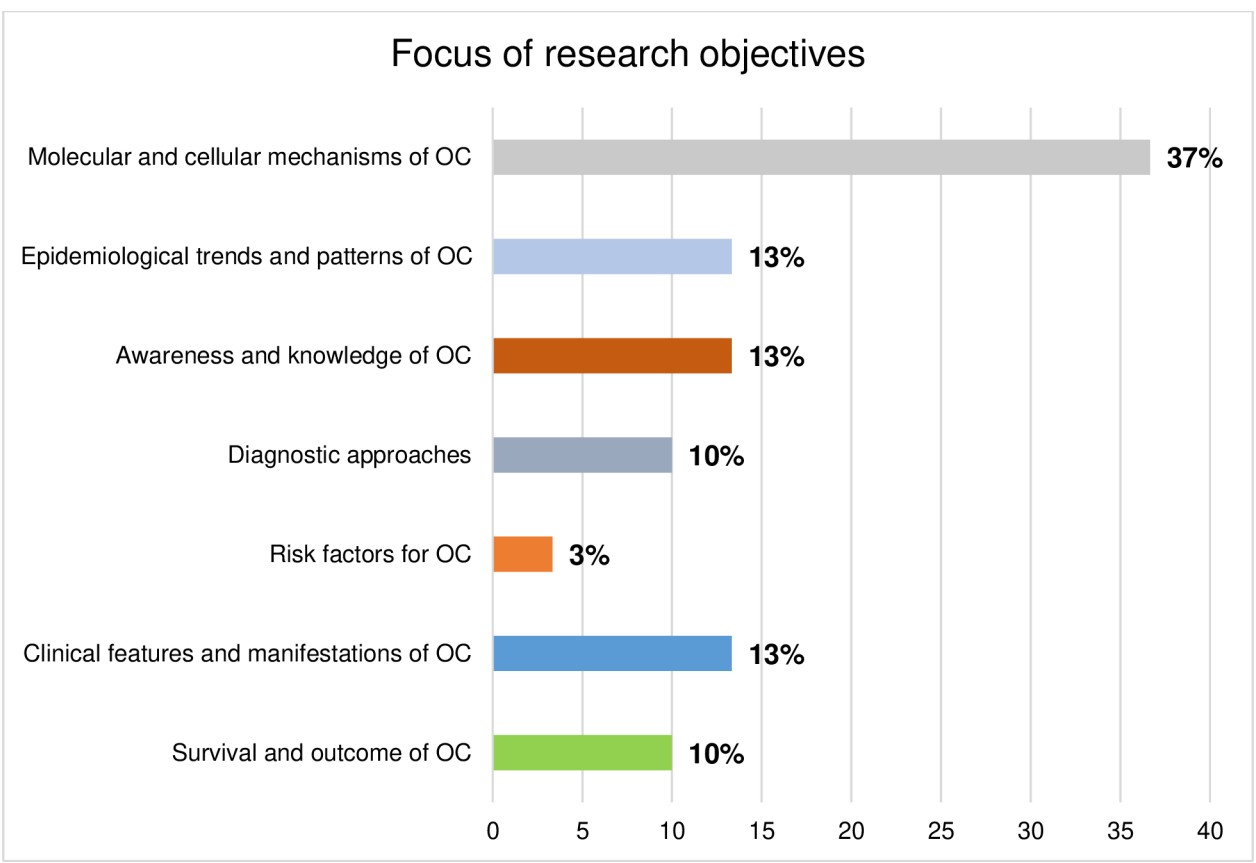

**Fig 6. Distribution of studies on OC in East Africa by research objectives.**

Some of these studies identified some risk factors for OC occurrence such as age [30,47], HIV status [30], and HPV infection [28]. In Sudan, HPV infection-related oral lesions showed significant association with cancer risk [28]. Likewise among Kenyans, HIV status was associated with OC incidence as majority (97%) of HIV patients were diagnosed with stage III/IV of malignant neoplasms and few (3%) had stage II [30].

In Kenya, Kaposi's sarcoma (KS), oral squamous cell carcinoma (SCC), and non-Hodgkin's lymphoma (NHL), were the most common malignant neoplasms which occurred in younger HIV patients than in the non-HIV group of patients [30]. Among Sudanese patients, OSCC was significantly associated with older age (older than 50 years) [47].

**Literacy on OC in East Africa.** Only few articles assessed public literacy on OC in East Africa. In Uganda, public literacy on OC was low (7%) in the community [45]. Comparatively higher proportion of community members (57.5%) [25] and dental patients (85.6%) in Sudan were aware of OC [29]. Media (66.1%) and health workers (33.9%) were primary information sources. Age influenced information sources as reported in the study [29].

In Sudan, majority of dental patients (57.7%) had good knowledge of OC signs and symptoms [29]. While another study in Sudan reported many members of the public as having moderate to high overall knowledge of OC (52.1%), good knowledge of OC risk factors (72.5%), symptoms (69.4%), and treatment and outcomes (64.2%) [25].

Similarly, dentists in Sudan had good knowledge of OC, such as the most frequent type of OC, oral site of infection, at risk group, risk factors, and early presentations of OC [24].

**Table 1. Summary of the studies on oral cancer in East Africa.**

| S/N | Author (Year) | Country of study | Study design | Study aim | Study setting | Study population | Sample size | Cancer type | Study instrument | Relevant findings | Study limitations/ conclusion |
|---|---|---|---|---|---|---|---|---|---|---|---|
| 1. | Anass & Ahmed (2013) [26] | Sudan | Case-control study | To assess oral epithelial proliferative activity by different cytological methods | Hospital | Healthy individuals and groups at high risk of oral cancer | 210 | Not specified | Buccal micronucleus assay Cytological proliferative marker methods | Toombak use was associated with cytological atypia (dyskaryosis) and keratinization. Use of Toombak predisposes users to infection and inflammatory conditions | Cytological proliferative marker methods should be standardised to provide rapid and effective oral cancer screening. |
| 2. | Sami et al., (2023) [48] | Sudan | Case-control study | To analyse the oral cavity microbial spatiality of the Sudanese and the implications that Toombak use brings post-OSCC development | Hospital | Toombak users and non-users | 78 | Oral squamous cell carcinoma | Puritan® Diagnostic Swabs, DNeasy Powersoil Pro Kit, | Oral cavity and keratinisation status is a better predictor of microbiome variation Toombak use. | Other behavioural factors that could indirectly influence the host-oral microbiome connections were not examined and require further research in the study population. |
| 3. | Dimba et al., (2007) [31] | Kenya | Cohort study (retrospective) | To describe oral diseases diagnosed in an urban referral centre in Kenya | Hospital | Patients who presented at diagnostic services | 1018 | Not specified | Not specified | Frequencies of oral tumours was high. Most cases of ameloblastomas and OSCCs were detected in younger patients. | The study reported incomplete socio-demographic data and failed to record clinical information on disease sites and stages of oral cancer. |
| 4. | Mohamed et al., (2021) [43] | Sudan | Cohort study (prospective) | To characterize the salivary mycobiome of oral squamous cell carcinoma (OSCC) patients and to explore patterns of diversities associated with overall survival (OS) | Hospital | Patients diagnosed with OSCC and healthy volunteers | 72 | Oral squamous cell carcinoma | FastDNA™ Kit | There is an association between *Candida* and poor prognosis of oral cancer. While the presence of *Malassezia* is an independent predictor of better prognosis. | Smaller sample size. Data on antibiotic use which tend to affect results of mycobiome analysis were not recorded. |
| 5. | Loro et al., (2000) [42] | Sudan and Norway | Cross-sectional analytical study | To examine apoptosis, expression of bax, bcl-2, and Ki-67 in OSCC | Hospital | Toombak users and non-users | 43 | Oral squamous cell carcinoma | TUNEL method | Apoptosis, bax, bcl-2, and Ki-67 in OSCC among Sudanese sample did not have any association with toombak use or p53 gene. | |
| 6. | Gaafar et al., (2022) [35] | Sudan | Cohort study (prospective) | To investigate the immune cell infiltrate in OSCC by using immunohistochemistry (IHC) for a panel of inflammatory cells in stromal and epithelial tumor compartments | Hospital | Patients diagnosed with primary OSCC and non-cancer patients | 45 | Oral squamous cell carcinoma | Image Scope software | Unlike in normal oral mucosal, cases of oral squamous cell carcinoma revealed a higher inflammatory infiltrate in the associated stroma. | Small sample size and age and gender difference observed between the case and control groups |

*(Continued)*

**Table 1.** (Continued)

| S/N | Author (Year) | Country of study | Study design | Study aim | Study setting | Study population | Sample size | Cancer type | Study instrument | Relevant findings | Study limitations/ conclusion |
|---|---|---|---|---|---|---|---|---|---|---|---|
| 7. | Okumu et al., (2012) [46] | Kenya | Cross-sectional analytical study | To evaluate the clinical features and histopathological types of orofacial malignant neoplasms in children. | Hospital | Patients not older than 15 years | 65 | Not specified | Questionnaire and clinical examination chart | Cases of neoplasms were more prevalent in children aged below 10 years. BL was more prevalent in boys than girls | Result not generalisable to reflect national prevalence of neoplasms. To reduce morbidity and mortality associated with neoplasms, reasons for late presentation of patients should be identified and addressed |
| 8. | Eltohami & Suleiman, (2023) [34] | Sudan | Cohort study (prospective) | To investigate the different clinical presentations of mucosal changes in WFC associated with OSCC | Hospital | Patients of OSCC diagnosed and surgically treated | 93 | Oral squamous cell carcinoma | Not indicated | OSCC patients present with advanced stage lesions which are connected to WFC, mostly among the Toombak users. | Sample selected only from the clinic and involved only surgically treated patients. |
| 9. | Elimairi et al., (2017) [32] | Sudan | Cross-sectional analytical study | To assess the clinical relevance of vital Lugol's iodine staining in diagnosing oral cancer, dysplastic lesions, and demarcation of the lesions' extent. | Hospital | Patients with suspicious oral epithelial lesions | 28 | Not specified | Vital Lugol's iodine staining | The study established the clinical relevance of vital Lugol's iodine staining in diagnosing oral cancer and dysplasia to be high | The investigation was conducted using a small sample of patients attending just one health facility. |
| 10. | Jalouli et al., (2011) [41] | Sudan | Case-control study | To examine the differential expression of nine selected genes related to apoptosis, cell cycle regulation, and intermediate filament proteins. | Hospital | Patients with OSCCs and their matched normal controls | 26 | Oral squamous cell carcinoma | TRIzol® reagent, Tissue RNA kit | More down-regulation of keratin 1 and keratin 13 was found in the OSCC of toombak users and non-toombak users than the control group. Levels of keratin 14 and keratin 19 and p16INK4a and p21WAF1/CIP1 in OSCC did not differentiate between users and non-users of toombak. | Not indicated |
| 11. | Gaafar, Osman, Elsheikh, et al, (2022) [36] | Sudan | Cohort study (prospective) | To evaluate patterns of immune cell infiltration at the ITF in a cohort of OSCC patients | Hospital | OSCC patients | 22 | Oral squamous cell carcinoma | Not indicated | PD-L1 expression within the invasive tumor front (ITF) epithelium independently predicts prognosis in oral squamous cell carcinoma (OSCC). CD20+ intraepithelial infiltration at the ITF is associated with advanced tumor stage. | Patients' cohorts used in the study had late OSCC stages. Small sample size was investigated. Failed to investigate the relationship between B-cell infiltrates with OSCC progression and outcomes. |

*(Continued)*

**Table 1.** (Continued)

| S/N | Author (Year) | Country of study | Study design | Study aim | Study setting | Study population | Sample size | Cancer type | Study instrument | Relevant findings | Study limitations/ conclusion |
|---|---|---|---|---|---|---|---|---|---|---|---|
| 12. | Ndayisabye et al., (2022) [6] | Rwanda | Cross-sectional analytical study | To determine factors associated with the OC adverse outcome and death | Hospital | Patients presenting diagnosis of oral cancer | 311 | Not indicated | Paper based checklist | OC adverse outcome and death were determined by education, poor oral hygiene, and religion. | Only a small sample of patients with stage-four OC were investigated. Due to the study design, factors associated with oral cancer such as the anatomical oral cancer sites were not examined. Secondary information was used. |
| 13. | Mohamed, et al., (2021) [44] | Sudan | Cohort study (prospective) | To test and validate the feasibility of e-nose technology for detecting OSCC in the limited-resource settings | Hospital | Patients with histologically confirmed primary OSCC and healthy non-cancer controls | 84 | Oral squamous cell carcinoma | E-nose devices | The overall diagnostic accuracy of the e-nose devices was high (81%) with a sensitivity and specificity of 88% and 71%, respectively. | The study involved small sample size. |
| 14. | Ibrahim et al., (2003) [39] | Sudan and Norway | Cohort study (prospective) | To identify potential differences in gene expression profile between OSCCs and matching normal oral mucosal tissues | Hospital | Patients with previously untreated OSCCs operated on | 22 | oral squamous cell carcinomas | 1987 UICC staging system | OSCCs-specific gene expression profile from both countries are well-defined. That is, alterations of 123 genes are common. There were no statistically significant differences observed in the expression of K13, 14, 19, and Bax between Sudanese and Norwegian OSCCs. | The expression and mutation status of identified genes were not compared to show their role in oncogenesis of OSCCs. |
| 15. | Ginawi et al., (2012) [37] | Sudan | Cohort study (retrospective) | To screen patients with oral lesions for the presence of Human Papilloma Virus (HPV) types 16 and 18. | Hospital | Patients with OSCC and those with oral benign lesions | 60 | Oral Squamous Cell Carcinomas | Immunohistological techniques | HPV types 16 and 18 were equally distributed among male and female patients. Majority of patients (66%) had well-differentiated carcinomas | Method used for the detection of HPV (sensitivity measures) may have insufficient sensitivity and low predictable value. Lack of patients' demographic and behavioral information |
| 16. | Ahmed & Mahgoob, (2007) [23] | Sudan | Cross-sectional descriptive study | To determine whether OC is gender-specific due to increased Toombak use among males as suggested a risk for subsequent development of oral cancer | Hospital | Patients with oral cancerous lesions and cases without apparent oral lesions | 82 | Oral Squamous Cell Carcinomas | Not indicated | Atypia was diagnosed only in male. Majority of patients with oral precancerous and cancerous lesions (70%) used toombak | Other risk factors of OC aside from toombak use were not assessed. |

*(Continued)*

**Table 1.** (Continued)

| S/N | Author (Year) | Country of study | Study design | Study aim | Study setting | Study population | Sample size | Cancer type | Study instrument | Relevant findings | Study limitations/ conclusion |
|---|---|---|---|---|---|---|---|---|---|---|---|
| 17. | Ibrahim et al., (2002) [38] | Sudan, Scandinavia, USA, and UK | Cross-sectional analytical study | To analyze the prevalence of mutations in exon 2 of the p21$^{wafl}$ gene of toombak dippers | Hospital | Tobacco smokers and non-smokers | 90 | Oral squamous cell carcinoma | PCR and direct DNA sequencing | Among Sudanese sample (smokers and non-smokers), no mutations were observed in the non-malignant oral mucosal lesions examined. | Did not analyse the role of p21$^{wafl}$ gene mutation in malignant changes among at risk groups. |
| 18. | Onyango et al., (2004) [5] | Kenya | Cohort study (retrospective) | To determine changes in the pattern of oral cancer in the past 20 years | Hospital | Cases of | 821 | Oral squamous cell carcinoma | Not indicated | All time prevalence of OC was low (3.6%). Authors observed a gradual decrease in cancers diagnosed at Kenyatta National Hospital. | Analysis of ethnic distribution was not conducted due to paucity of reliable information. |
| 19. | Babiker et al., (2017) [29] | Sudan | Cross-sectional analytical study | To assess oral cancer awareness regarding knowledge of signs, symptoms, risk factors and sources of the information | Hospital | Adult dental patients | 500 | Not indicated | Interviewer-administered questionnaire | Majority (57.7%) had good knowledge of signs and symptoms of OC. Media (66.1%) and health workers (33.9%) were the main sources of information. Despite expressing good attitude towards cancer screening, rate of screening uptake was low. | The study relied on interview reporting which is prone to recall bias and information bias. Sample selected from patients attending dental clinic alone and are not generalisable |
| 20. | Ahmed & Naidoo, (2019) [24] | Sudan | Cross-sectional analytical study | To determine the knowledge, attitude and practice of dentists related to oral cancer prevention and early detection in public dental clinics | Hospital | Dentists | 130 | Not indicated | Structured questionnaire | Aware of the major risk factors and common presentations of oral cancer was high. Majority desired further training on oral cancer | Sample selected strictly among dentists only |
| 21. | Sand et al., (2012) [49] | Sudan | Cohort study (prospective) | To examine the prevalence of p53 codon 72 polymorphism in brush biopsies and relate the findings toombak use and high prevalence of OSCC in Sudan. | Hospital | Toombak consumers or non toombak-using volunteers | 174 | Oral squamous cell carcinoma | Polymerase chain reaction/ restriction fragment length polymorphism | Arg/Arg (18%), Pro/Pro (29%) and Arg/Pro (52%) genotypes were higher among toombak users than non-users | Did not examine the role of p53 polymorphism in human oral cancers. |
| 22. | Butt et al., (2008) [30] | Kenya | Cross sectional descriptive study | To determine the pattern of occurrence of head and neck malignancy among HIV-infected patients | Hospital | HIV-infected patients with neoplastic and non-neoplastic lesions | 200 | HIV-associated neoplasms | Haematoxylin and eosin staining. | Among patients with neoplastic lesions (27%); KS (68%); SCC (17%), NHL (13%); and Burkitt's lymphoma (2%) are the prevalent lesions. KS and SCC lesions were common among females. | Focus only on HIV patients |

*(Continued)*

**Table 1.** (Continued)

| S/N | Author (Year) | Country of study | Study design | Study aim | Study setting | Study population | Sample size | Cancer type | Study instrument | Relevant findings | Study limitations/ conclusion |
|---|---|---|---|---|---|---|---|---|---|---|---|
| 23. | Osman et al., (2010) [47] | Sudan | Cross sectional descriptive study | To describe the pattern of cancer cases attending a referral oral and maxillo-facial hospital in Sudan during the period 2006–2007 | Hospital | All cancer cases registered during the study period | 261 | Not specified | Not indicated | Intraoral squamous cell carcinoma (73.6%) was the most common type of cancer. Patients present at the advanced stage of the tumor. | All participants were interviewed in the hospital during treatment |
| 24. | Nabirye & Kamulegeya, (2019) [45] | Uganda | Cross sectional descriptive study | To assess the levels of awareness and knowledge about oral cancer, its causes and or risk factors among Ugandan patients seeking oral healthcare | Community | Adult patients who attended a free dental camp | 188 | Not specified | Assistant-administered questionnaire | Level of awareness/ knowledge about oral cancer was low. Uptake of oral cancer screening was low | Reliance on self-reports in the presence of health workers. Sample used not representative of the target population |
| 25. | Al-Hakimi et al., (2016) [25] | Sudan | Cross-sectional analytical study | To assess the current level of public knowledge of oral cancer and effects of demographic factors on knowledge | Hospital | Outpatients at major hospitals | 501 | Not specified | pretested structured questionnaire | About half of the sample were aware of OC. Knowledge of OC varied. age and place of residence are significant determinants of oral cancer knowledge, | Study's scope is constrained by its cross-sectional design which does not support model development. |
| 26. | Eltohami & Sulaiman, (2023) [33] | Sudan | Cohort study (prospective) | To investigate the clinicopathological behaviour of OSCC and predictors of development of recurrence in patients. | Hospital | OSCC patients surgically treated | 93 | Oral squamous cell carcinoma | Fine needle aspiration cytology and computed tomography with contrast | Majority (82%) were diagnosed as stage IV tumors and 62.4% were nodal metastasis. Local recurrence of OC was associated with toombak dipping, tumor staging, and mucosal field changes. | |
| 27. | Babiker et al., (2013) [28] | Sudan | Cohort study (retrospective) | To identify and genotype the HR-HPV subtypes in oral tissues obtained from Sudanese patients with oral lesions | Hospital | patients with oral lesions | 200 | Not specified | Molecular methods (PCR) | Oral cancer incidence increased with age, predominantly affecting males (62%) compared to females (38%). HPV infection, detected in 6% of oral lesions, showed significant association with cancer risk | Samples not analyzed concurrently with normal oral tissues or tumours. No information available regarding patients' socio-economic status, nutritional status, health history, or family relations and alcohol intake and smoking habits. |
| 28. | Ahmed et al., (2003) [22] | Sudan | Cohort study (retrospective) | To evaluate the role of exfoliative cytology in the assessment of the presence and severity of oral epithelia atypia among toombak dippers and tobacco smokers | Community | Toombak dippers; cigarette smokers; non-tobacco users and OSCC patients | 300 | Precancerous/ premalignant lesions | Exfoliative cytology (EFC) | Epithelia atypia was detected in 29 (10%) individuals, majority of them smoked tobacco (48%), toombak (38%), and 14% did not smoke at all. All OSCC patients had epithelia atypia | Did not evaluate the role of toombak and tobacco in the occurrence of epithelia atypia in buccal mucosa |

*(Continued)*

**Table 1.** (Continued)

| S/N | Author (Year) | Country of study | Study design | Study aim | Study setting | Study population | Sample size | Cancer type | Study instrument | Relevant findings | Study limitations/ conclusion |
|---|---|---|---|---|---|---|---|---|---|---|---|
| 29. | Asio et al., (2018) [27] | Uganda | Cohort study (retrospective) | To establish survival and associated factors among patients with oral squamous cell carcinoma | Hospital | Histologically confirmed oral squamous cell carcinoma (OSCC) patients | 384 | Oral squamous cell carcinoma | Kaplan-Meier method | Patients' survival was determined by clinical stage, poorly differentiated histo-pathological grade, gender (male), age (> 55 years) at diagnosis and moderately differentiated histo-pathological grade. Use of tobacco and alcohol, tumour location and treatment group did not determine survival. | In addition to being a hospital-based study, the research outcome was hindered by the loss of many patients to follow up. Participation of the said patients could have produced a different result on survival of OSCC. |
| 30. | Ibrahim et al., (2005) [40] | Sudan and Sri Lanka | Cohort study (prospective) | Proteomic methodologies were utilized to analyze protein alterations in oral squamous cell carcinomas (OSCCs) originating from Sudan and Sri Lanka, with the aim of identifying shared biomarkers in OSCCs across both nations. | Hospital | Patients with OSCC and healthy controls | 56 | Oral squamous cell carcinoma | Antibody microarrays, two-dimensional electrophoresis and matrix-assisted laser desorption/ionization time-of-flight mass spectrometry (2-DE/ MALDI-TOF-MS) | Abundance level of protein was present in tumour than normal samples. A limited number of proteins were discovered to be either over-regulated (4) or under-regulated (6) in OSCCs. | Defining major cellular changes in the transition from normal to malignant oral cancer is hindered by defects in the diversity of 2-DE features. |

However, majority of the dentists lacked knowledge regarding the survival rate and low consumption of fruits and vegetables as a risk factor for OC [24].

Despite Ugandan participants having low awareness, their knowledge of risk factors for OC was better as they mentioned smoking (66.5%) alcohol intake (44.9%), oral sex (34.1%), sun exposure (11.4%), inadequate fresh fruits consumption (11.4%), HIV status (35.7%), and older age (23.2%) as risk factors [45]. In Sudan, dental patients identified common signs and symptoms of OC to include unhealed ulcers (67.2%), change in colour (65%), white patches (63.6%), and soreness (62.4%) [29].

Studies conducted in Uganda [45] and Sudan [24,25,29] found knowledge of OC to be influenced by age, gender, education, place of residence and attitude towards OC screening among others. In Sudan, those under 40 years were more likely to report ulcers (70%), while female participants who had positive attitude towards OC screening recognised colour changes (69%), and white patches (69.5%) as signs [29].

In Sudan, the overall knowledge of OC was associated with age, place of residence and educational level [25]. Additionally, among dentists practicing in Sudan, knowledge of OC varied by gender, while time of graduation impacted knowledge of the most common OC type [24].

**Risk factors of oral cancer in East Africa.** Toombak use was reported as a major behavioural risk factor for OC, especially among Sudanese, and majority of the participants in these studies identified as toombak users [22,23,26,33,34,41,42,48,49]. Many of the studies also found significant association between toombak or tobacco use and diagnosis of OC. Only

a few studies did not find an association [41,42]. Furthermore, some of the studies discovered some relationships between toombak use and severity of oral diseases [22].

Toombak use was significantly linked to overlapping lesions of the mouth (56%) and occurrence of OSCC (88%) in patients [47]. Likewise, cytological changes, including alteration of oral microbiome [48], atypia, keratinization, increased nucleolar organizer region (NOR) counts, micronuclei frequencies, susceptibility to infections, and inflammation [26] are all attributed to toombak use. NOR counts and micronuclei frequencies were significantly associated with keratinization, duration, and frequency of Toombak use [26].

In addition to toombak use, other major behaviours linked to the occurrence of OC are tobacco smoking, alcohol drinking and a combination of more than one of the three [35,47]. In a different study, cytological atypia was diagnosed among individuals over 36 years old who had been using Toombak for more than 21 years. These findings underscore the potential health risks associated with Toombak consumption, particularly in relation to oral health and cytological abnormalities [26].

In addition to the above poor oral hygiene was also another behavioural risk factors; a higher proportion of some investigated East African (Sudanese) patients with OSCC stated that their oral hygiene practice was poor [35].

**Diagnosis of oral cancer in East Africa.** Diverse oral cancer types have been reported in East Africa. Fig 7 depicts these cancer types. Additionally, this figure shows statistics of those articles that also reported oral precancer.

Some of the reviewed studies assessed the validity of instruments and devices designed for OC diagnosis. One of the studies tested the feasibility of a portable electronic nose for detection of OSCC. The overall diagnostic accuracy of the e-nose devices was high (81%) with a sensitivity and specificity of 88% and 71%, respectively [44]. Another study tested the clinical relevance of vital Lugol's iodine staining in OC and dysplastic lesions diagnosis. Its clinical relevance level was found to be high (90.0%) and it was described as easy, safe, effective and of great value in recognising and diagnosis of OC and dysplasia [32].

Pertinently, most OC cases across East Africa were purportedly presented and diagnosed late. In Sudan, 76.3% of patients who had OC tumours [34], and 85.8% of OSCC patients presented at the advanced stage [44,47]. While in Kenya, 53.5% of HIV patients with malignant neoplasms [30], and majority (61%) of the OSCC samples in Uganda [27] were already advanced. Some of the tumours were size T4a [34], some in TNM stage III and IV [27,30,35,47].

A study also found that the majority of dentists in Sudan did not carry out special examinations to diagnose OC for reasons, such as inadequate training, belief that special examinations are unnecessary for all patients and because it is time-consuming. Many (52.2%) had made referral of suspected cases to specialists. However, majority (66.4%) believed that they were untrained to detect oral cancer lesions, hence most dentists perceived their knowledge as out-of-date while some believed they lacked adequate knowledge of OC prevention, and diagnosis. Almost all participants (95.6%) were desirous of additional training on oral cancer [24].

**Clinical status of diagnosed oral cancer cases in East Africa.** *Anatomical sites:* OC lesions were identified in either single or multiple sites in the oral cavity. Almost all tumours (99%) in one of the studies were found in multiple sites [34]. Common sites of OC in East Africa include the tongue [5,23,27,31,32,41,47], floor of the mouth [5,27,31,32,41,47], buccal mucosa [23,27,31,32,35,41], palate [27,31,41] and overlapping lesion of the mouth [47].

OC malignancies were likewise detected in other locations, such as mandible, maxilla [5,31], gingival [23,41], maxillary mucosa, retromolar region and upper labial vestibule [41], gingivolabial mucosa [34], lip, gum, and unspecified location in the mouth [47]. Others are

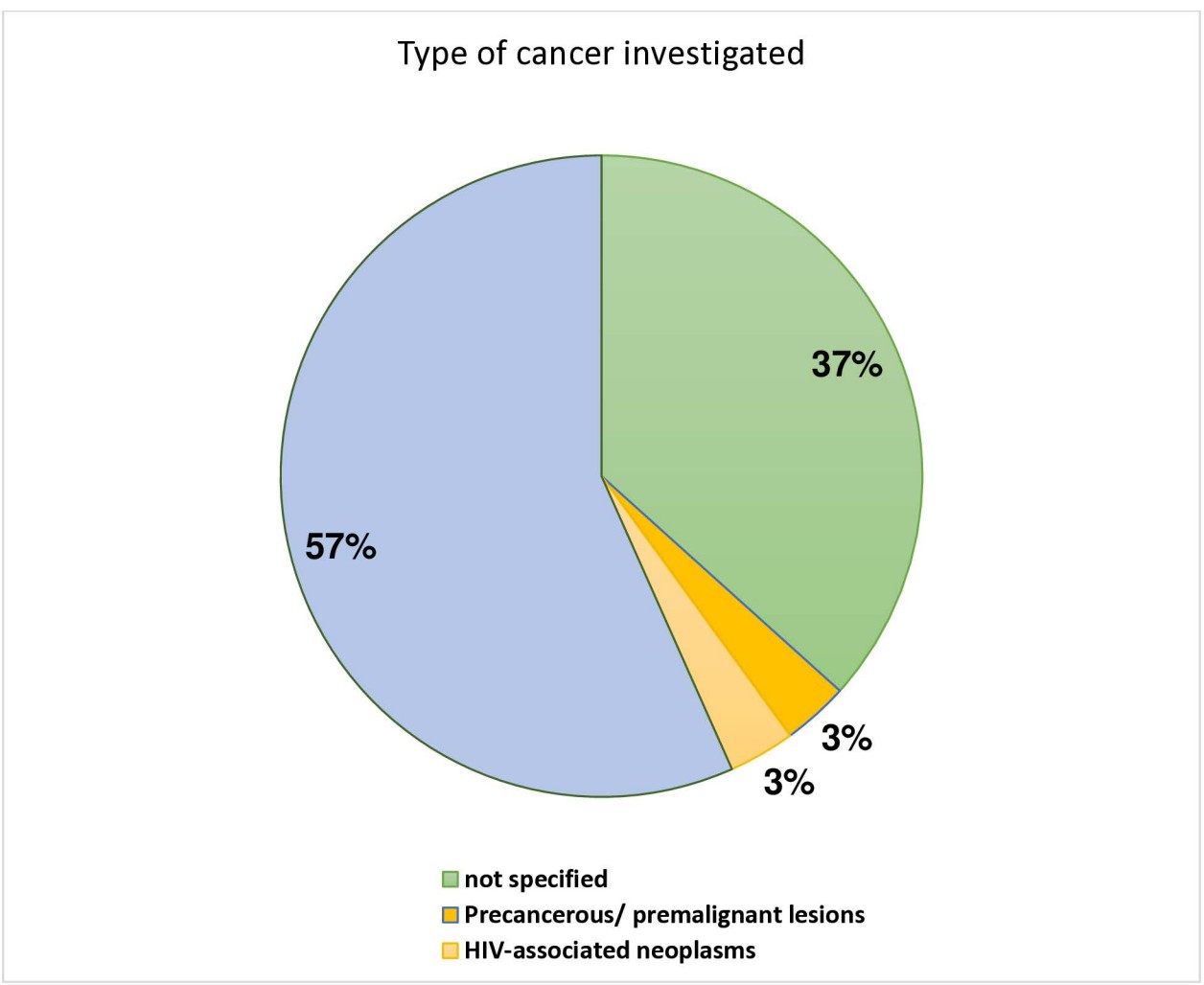

**Fig 7. Distribution of included articles by type of cancer investigated.**

upper labial mucosa [32,35], ventral surface of the tongue, the lower labial mucosa [32], and sulcular mucosa [35].

In Sudan, lesions were most often detected in sites such as gingivobuccal mucosa (74.2%) [34], overlapping lesion of the mouth [47], labial sulcus (23%) [41]; lower lip (42.8%/ 54.2%) [5,23,32]; and the tongue [5]. In Uganda, the tongue (34.1%) was the most common site [27]. Likewise, tongue was the most affected site in Kenya [5]. Also, in Kenya, most benign tumours originated in the mandible region [31].

The nature/type of the neoplasms also determines their sites in the oral cavity. Most Kaposi's sarcoma (KS) lesions were identified on the palate and the maxillary alveolus and gingivae, while non-Hodgkin's lymphoma (NHL) and OSCC were found mainly in the maxillary and mandibular alveolus [30].

***Histopathological groupings*:** Majority of the OC tumours and lesions identified in patients in the reviewed studies were categorised as well-defined or well-differentiated carcinomas in 51.6%, 33.4%, 66%, and 88% of the samples [27,35,37,41]. Other carcinomas were classified as poorly defined (21.9%) [27], and moderately defined (12%) [41].

Only one study, from Kenya, investigated various histopathological types of OC, and this report was on children [46]. Common diagnosis of OC in children were burkitt's lymphoma (BL) (21%), rhabdomyosarcoma (RMS) (13%), non-Burkitt's non-Hodgkin's lymphoma (NB-NHL) (9%), osteosarcoma (OS) (4%), Kaposi's sarcoma (KS) (4%), squamous cell carcinoma (SCC) (3%), fibrosarcoma (FS) (1%), nasopharyngeal carcinoma (NPC) (1%), in their respective order. All of which were primary tumours and no obvious clinical evidence of distant metastasis was noticeable [46].

*Genetic and proteomic compositions*: Arg/Pro heterozygote was dominant (53%) in a study. Differences in Arg/Arg and Pro/Pro proportions between chronic toombak users and non-users were statistically significant (ref). While no significant difference was observed for the Arg/Pro genotype. Pro/Pro homozygosity was associated with toombak use as it was exclusively found in chronic toombak users. Likewise, Arg/Arg (18%) and Arg/Pro (52%) genotypes were higher among toombak users than non-users [49].

OSCC from toombak-dippers showed a higher prevalence of p21waf1 exon 2 mutations compared to non-snuff-dippers. OSCCs from both groups exhibited mutations in p21waf1 and p53 genes, with some OSCCs also showing S100A4 gene mutations. The findings suggest that p21waf1 mutations may be linked to oral carcinogenesis in toombak-dippers, indicating the potential role of tobacco-specific nitrosamines. Additionally, mutations in S100A4 may complement p21waf1 and p53 mutations in OSCCs [38].

Protein distribution in OC tumours varied, and the level abnormally distributed in some of the studies. In a multi-country study, an abundance level of protein was present in OC tumour than normal samples. Eight proteins (NEK2, p56dok2, Gap1m, neuroglycan C, pp120 src substrate, CAF-1 p150, EBP50, PMF-1) implicated in crucial cellular functions were documented. In Sudanese OSCC sample, CAF-1 p150 and Adenovirus 5E1A exhibited the most significant fold changes. In the same study, a limited number of proteins were discovered to be either over-regulated (4) or under-regulated (6) in OSCCs [40].

Comparatively, in Sudanese patients, proteins such as Bcl2, keratin 1 and keratin 13, were all downregulated in the OSCC of toombak users and non-toombak users than the control group. Low expression of keratin 1 and keratin 13 was also recorded in non-users than the control group. Similarly, survivin was expressed in the OSCC of the toombak users at a high level than the control. P53 was also down-regulated in the toombak users OSCC samples compared to their normal controls suggesting that tumour cells are unable to synthesize keratin 1 and 13 [41]. In another study, no significant differences were noted in apoptosis, Bax, Bcl-2, and Ki-67 expression in Sudanese OSCC cases based on toombak use or p53 gene status [42]. Furthermore, apoptosis correlated with Bax expression and remained unaffected by p53 gene status or toombak use [42].

Tumour samples from Sudan exhibited altered expression of S100A7, S100A9, glutathione transferase, lactoglobulin, and stratifin, while normal controls compared to tumours showed changes in ferritin light subunit, fast skeletal myosin, fibrinogen, and a hypothetical protein. Proteins associated with tumours, such as psoriasin, calgranulin-B, and glutathione transferase, exhibited alterations in oral cancers compared to normal controls [40].

OSCCs-specific gene expression profile from both countries are well-defined. That is, alterations of 123 genes are common. There were no statistically significant differences observed in the expression of K13, 14, 19, and Bax between Sudanese and Norwegian OSCCs [39].

*Treatment outcomes*: Only five studies reported findings associated with OC treatment/prognosis in East Africa [6,27,33,36,43]. Two of these studies examined the influence of microbiomes profile on survival [36,43], two studies explored related clinical and social factors [6,27], while the rest investigated recurrence of OC [34].

In Sudan, PD-L1 expression within the invasive tumour front (ITF) epithelium independently predicts prognosis in oral squamous cell carcinoma (OSCC) [36]. Another study

found an association between *Candida* and poor prognosis of OC, while the presence of *Malassezia* was an independent predictor of better prognosis/survival [43].

Several key factors predictive of local and regional recurrence in oral squamous cell carcinoma (OSCC) were identified among Sudanese patients. Specifically, Toombak dipping, tumour staging, and the presence of mucosal field changes were significant predictors for local recurrence. Also, positive pathological lymph nodes and extranodal extension were significant predictors for regional recurrence. However, there were no significant associations found between local and locoregional recurrences and lympho-vascular invasion or perineural invasion [33].

Factors like level of education, province of residence, oral hygiene, and religion significantly contributed to OC adverse outcome and death in Rwanda [6]. Whereas in Uganda, patients' survival was determined by clinical stage, poorly differentiated histo-pathological grade, gender (male), age (> 55 years) at diagnosis and moderately differentiated histo-pathological grade. Patients diagnosed with OSCC exhibited a low survival rate of 20.7% after five years, with nearly half of them (43.6%) succumbing within two years of diagnosis [27].

However, age, gender, marital status, primary site of infection, and wealth index, did not have significant influence on OC outcome in Rwanda [6]. Furthermore, in Uganda, tobacco and alcohol consumption, tumour location and treatment group did not determine survival [27].

The overall death rate of 27.81 per 100 person-years was observed in a Ugandan sample. Chances of two-year survival was higher (43.6%) than five-year survival (20.7%). Five-year survival rate for tumour found on the lip was the best (100%) while tumours arising from floor of the mouth and alveolus had no survivor at 5 years [27].

## Discussion

Reviewing the existing evidence of oral cancer awareness, burden and outcomes in east Africa is crucial to improving the prevention, diagnosis, and treatment of cancer especially since there is a predicted rise in disease incidence and mortality [50]. This comprehensive review examined the nature and extent of empirical findings available in existing literature on oral cancer in East African countries with emerging themes covering epidemiology, risk factors, and clinical characteristics of OC across East Africa. The status of evidence for our research question is comparable to other regions in Africa and even worldwide, however there are findings which appear distinct to East Africa.

### Epidemiology

This review found evidence to support age and gender disparities in OC, with its incidence significantly increasing with age and predominantly affecting males. These findings substantiate the results from previous studies that incidence of OC increases with age [51,52]. Other research also revealed gender difference in the occurrence of OC [53] and other forms of cancer [54]. Cumulative stress and damage to cells over time is associated with transition to cancer and is mostly the basis to explain the increased risk of cancer with ageing. The higher likelihood of males engaging in oral cancer risks behaviours is a possible reason why they have a dominance of oral cancer incidence. Age and gender disparities in OC is found in most regions of the world and presentations vary based on etiological factors.

This review found a recent increase in the publication of oral cancer cases which is in conformity with global trends and is due to several factors including: rising burden of oral cancer, increasing international research collaborations on cancer, increasing thematic areas of oral cancer research and emerging important etiological factors of oral cancer [55]. However, most of the reports in this review of East Africa OC were from Sudan, and although the reasons for

this are not very clear, it may be due to some of the factors stimulating global OC publications and possibly interests or capacity of existing faculty.

## Risk factors

In this review, toombak use alone or toombak and tobacco use emerged as significant behavioral risk factors for oral cancer, particularly among Sudanese populations [56,57]. Toombak use was linked to the severity of oral diseases. However, some of the reviewed studies did not find this association. These findings underscore the importance of early detection strategies and targeted interventions tailored to the unique epidemiological and clinical profiles of oral and maxillofacial tumors in East Africa. Public health interventions aimed at curbing the prevalence of Toombak use and promoting healthier lifestyle choices to reduce the incidence of oral diseases in Sudan and other regions where these behaviors are prevalent cannot be overemphasized.

Some of the studies, in this review, revealed microbiome differences in individuals at risk or diagnosed with OC compared to controls. In Sudanese patients, oral microbiomes including bacteria and fungi like *Saccharomyces, Aspergillus,* and *Candida*, were common with some variations in different patient samples. OC patients in particular had unique microbiomes, including Candida and Saccharomyces. This aligns with previous studies that identified diversity of oral microbiomes in OC samples [58–60]. Some of the microbiomes identified in the review have also been reported in other reports [61]. Although the importance of oral microbiomes in OC is well-established [62,63], most studies that have investigated oral microbiomes in cancer have focused on cancer in general or cancer of other body organs [63].

In the current review, gender and age influenced microbiome presence, while smoking status affected fungal and bacterial species in oral sites. Core microbiome analysis showed significant differences between Toombak users and non-users across mucosal locations. Toombak users had distinct microbiomes, with *Streptococcaceae, Blumeria*, and *Issatchenkia* being more prevalent. *Candida* presence correlated with poorer OC prognosis, while *Malassezia* presence suggested a better prognosis [61], indicating oral microbiome's diverse role in OC risks. In keeping with our review, several studies have also reported that sociodemographic differences like race, gender, education, diet, smoking and oral health behaviors influences the composition of the oral microbiome in persons with or without oral diseases [64,65].

## Health behaviors

Health seeking for OC and related oral diseases in East Africa is often characterised by late clinical presentation and advanced-stage diagnosis (III and IV), particularly in Sudan, Kenya, and Uganda. Many studies have reported delayed health seeking for OC as a common experience [66–68]. Some of the factors found to be responsible include male gender, low-skill occupations, smoking behaviour, self-detection of symptoms [68], personal beliefs about symptoms, interpretation of diagnosis, limited access to health professionals, patient's social responsibilities [67], patient's knowledge, their anxiety levels, and nature of their condition. Additionally, healthcare provider-related factors, such as the experience of general practitioners (GPs), delays in referrals, and the perception of younger age groups as low-risk, contribute to symptom oversight [69]. Kolude et al reported that in a West African tertiary hospital, a combination of patients and professional delay negatively influenced the management of oral cancer patients but the patient's factors formed the bulk of the delay [70]. However, clinical presentation for pediatric cancer was earlier than other age group cancers. Also, for other age groups symptoms like bleeding prompted quicker treatment seeking. Survival in OC is significantly influenced by the timing of diagnosis as early stage cancers show a survival

rate of over 90%, whereas it drops to 5–20% in stage III and IV disease, therefore, early diagnosis is crucial in improving patient's prognosis [71].

## Prognosis

This current scoping review reveals a multifaceted interplay of prognostic factors, although only few studies in the region investigated these factors. Factors which were focused on include microbiome profiles, clinical and social determinants, and recurrence patterns in OC. Notably, this review has elucidated the prognostic relevance of PD-L1 expression at the invasive tumor front and microbial composition, with *Candida* correlating with adverse prognosis and *Malassezia* showing associations with improved survival rates. Furthermore, factors such as Toombak usage, tumor staging, and mucosal field alterations have emerged as predictors of local recurrence, while lymph node involvement and extranodal extension have been associated with regional recurrence among Sudanese cohorts.

In Rwanda, education level, province of residence, oral hygiene practices, and religious affiliation were significant contributors to adverse OC outcomes and mortality. Conversely, age, gender, marital status, and socioeconomic status showed limited influence on OC prognosis in the same contexts. Similarly, in Uganda, survival rates were influenced by clinical stage, histopathological grade, age, and gender, with overall poor survival outcomes observed, particularly among patients diagnosed with oral squamous cell carcinoma (OSCC). Notably, lifestyle factors such as tobacco and alcohol consumption did not emerge as significant determinants of survival in Ugandan OC patients. These findings confirm the intricate nature of factors shaping OC outcomes in East Africa and underscores the importance of comprehensive prognostic evaluations that integrate clinical, social, and microbial dimensions. Such insights are vital for guiding targeted interventions aimed at enhancing OC management and improving patient survival across the region and by extension, Africa.

## Outcomes of review and future considerations

This scoping review shows gaps and discernible disparities in awareness and knowledge levels across East Africa. Dental patients and dentists generally exhibit heightened awareness and knowledge relative to the broader populace. Noteworthy negative influencers of knowledge include increasing age, male gender, low-level education, and rural residency status. However, gaps persist in knowledge regarding specific risk factors and preventive measures. For example, screening adherence remains lackluster, particularly among older demographics and individuals with lower educational attainment.

Furthermore, few studies undertook an evaluation of instrument validity for OC detection, revealing promising outcomes concerning diagnostic accuracy and instrument reliability. Hence, the imperative for targeted awareness initiatives and enhanced training of healthcare personnel to bolster OC detection and management endeavors in East Africa, specifically and Africa in general.

Based on the evidence presented, several potential areas for systematic review on OC in East Africa have been identified. First is the role of the oral microbiome in OC prognosis across East Africa. The current review observed variations in microbiome profiles among OC patients, certain bacteria and fungi, such as Candida and Saccharomyces, were linked to better or worse outcomes. A systematic review could explore these microbiome differences and their impact on OC progression, thereby potentially clarifying conflicting findings and contributing to more precise prognostic tools.

Second is the influence of behavioral, sociodemographic, and systemic factors on OC outcomes and health-seeking behaviors in East Africa. Behavioral factors like Toombak and tobacco use have been identified as significant risk factors, results are however inconsistent,

necessitating further investigation. Third, sociodemographic factors such as age, gender, and education affect awareness, treatment-seeking behaviors, and cancer outcomes, suggest the need for targeted public health interventions. Delayed health-seeking behaviors and late-stage diagnoses due to personal and systemic issues also call for a review of factors contributing to these delays. Finally, evaluating the effectiveness and reliability of diagnostic tools used in the region could improve detection practices. Addressing these areas through systematic reviews could lead to more effective interventions and improved patient outcomes in East Africa.

## Limitations

While the findings offer valuable insights, several potential limitations should be noted. First, the results are specific to East Africa and may not be applicable to other regions with different epidemiological profiles for oral cancer. As a scoping review, the study did not include detailed quantitative analysis, which limits the ability to draw specific conclusions. Additionally, the unique aspects of oral cancer in East Africa, such as the influence of Toombak use, may not be relevant elsewhere. The review also did not fully consider the cultural, socioeconomic, and behavioral factors that affect oral cancer prevalence and prevention strategies in the region. Furthermore, the findings did not account for the diversity within East Africa, including variations in health systems, cultural practices, socioeconomic conditions, and access to healthcare.

## Conclusion

This scoping review has mapped the existing empirical evidence and gaps concerning oral cancer in East Africa. Based on the review findings, it could be concluded that there is a dire need for targeted interventions and early detection strategies tailored to the unique epidemiological and clinical profiles of oral and maxillofacial tumors in East Africa. Public health interventions aimed at curbing the prevalence of Toombak use and promoting healthier lifestyle choices to reduce the oral diseases incidence in Sudan and other regions where these behaviors are prevalent remain germane.

## Supporting information

**S1 Table. Search strings used on PubMed database.**
(DOCX)

**S2 Table. Search strings used on SCOPUS database.**
(DOCX)

**S3 Table. Search strings used on AMED – The Allied and Complementary Medicine Database, APA PsycArticles, APA PsycInfo, CINAHL Ultimate, Dentistry & Oral Sciences Source, Psychology and Behavioral Sciences Collection, SPORTDiscus with Full Text databases (via EBSCOHost interface).**
(DOCX)

**S4 Table. List of literature considered for full text screening.**
(DOCX)

**S5 Table. Quality appraisal outcomes of the appraised quantitative randomized studies using the Mixed Methods Appraisal Tool.**
(DOCX)

**S6 Table. Quality appraisal outcomes of the appraised quantitative non-randomized studies using the Mixed Methods Appraisal Tool.**
(DOCX)

**S7 Table. Quality appraisal outcomes of the appraised quantitative descriptive study using the Mixed Methods Appraisal Tool.**
(DOCX)

**S8 Table. Quality appraisal outcomes of the appraised qualitative study using the Mixed Methods Appraisal Tool.**
(DOCX)

## Author contributions

**Conceptualization:** Kafayat Aminu, Timothy Olukunle Aladelusi, Akinyele Olumuyiwa Adisa, Afeez Abolarinwa Salami, Jimoh Amzat, Kehinde Kazeem Kanmodi.

**Data curation:** Chiamaka Norah Ezeagu, Afeez Abolarinwa Salami, Jacob Njideka Nwafor, Kehinde Kazeem Kanmodi.

**Formal analysis:** Chiamaka Norah Ezeagu, Afeez Abolarinwa Salami, Jacob Njideka Nwafor, Jimoh Amzat, Kehinde Kazeem Kanmodi.

**Methodology:** Chiamaka Norah Ezeagu, Afeez Abolarinwa Salami, Kehinde Kazeem Kanmodi.

**Project administration:** Afeez Abolarinwa Salami.

**Supervision:** Jimoh Amzat, Kehinde Kazeem Kanmodi.

**Writing – original draft:** Kafayat Aminu, Timothy Olukunle Aladelusi, Akinyele Olumuyiwa Adisa, Chiamaka Norah Ezeagu, Afeez Abolarinwa Salami, Jacob Njideka Nwafor, Jimoh Amzat, Kehinde Kazeem Kanmodi.

**Writing – review & editing:** Kafayat Aminu, Timothy Olukunle Aladelusi, Akinyele Olumuyiwa Adisa, Chiamaka Norah Ezeagu, Afeez Abolarinwa Salami, Jacob Njideka Nwafor, Peace Uwambaye, Jimoh Amzat, Julienne Murererehe, Semeeh Akinwale Omoleke, Mohammed Abdulaziz, Ruwan Duminda Jayasinghe, Kehinde Kazeem Kanmodi.

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
