## [Decision Letter · Decision Letter 0]

20 Aug 2024

PONE-D-24-22531ORAL CANCER IN EAST AFRICA: A SCOPING REVIEW OF EMPIRICAL EVIDENCEPLOS ONE

Dear Dr. Murererehe,

Thank you for submitting your manuscript to PLOS ONE. After careful consideration, we feel that it has merit but does not fully meet PLOS ONE’s publication criteria as it currently stands. Therefore, we invite you to submit a revised version of the manuscript that addresses the points raised during the review process.

We look forward to receiving your revised manuscript.

Kind regards,

John Adeoye

Academic Editor

PLOS ONE

Journal Requirements:

2. We note that your Data Availability Statement is currently as follows: "All relevant data are within the manuscript and its Supporting Information files".

4. We notice that your supplementary figure are included in the manuscript file. Please remove them and upload them with the file type 'Supporting Information'. Please ensure that each Supporting Information file has a legend listed in the manuscript after the references list.

Additional Editor Comments:

- Authors should address the reviewers comments in the attachment.

- Include a critical assessment of the individual sources of evidence to determine their quality and bias.

- Authors should comment (in the discussion) on the areas for potential systematic reviews on this topic in the future.

Reviewers' comments:

Reviewer's Responses to Questions

**Comments to the Author**

1. Is the manuscript technically sound, and do the data support the conclusions?

Reviewer #1: Yes

Reviewer #2: Partly

2. Has the statistical analysis been performed appropriately and rigorously? 

Reviewer #1: Yes

Reviewer #2: No

3. Have the authors made all data underlying the findings in their manuscript fully available?

Reviewer #1: Yes

Reviewer #2: No

4. Is the manuscript presented in an intelligible fashion and written in standard English?

Reviewer #1: Yes

Reviewer #2: Yes

5. Review Comments to the Author

Reviewer #1: The manuscript was well written, the methodology was very robust and result presentation highly scientific. Additionally, the process for reporting systematic reviews and meta-analysis was strictly followed.

Reviewer #2: Thank you for the opportunity to review your work. I commend the effort. See the attachment for my comments.

I would prefer you discuss your findings along major these such as risk factors/aetiology, pathogenesis, treatment, prognosis. it will make the work easier to read and cited.

6. PLOS authors have the option to publish the peer review history of their article (what does this mean? ). If published, this will include your full peer review and any attached files.

**Do you want your identity to be public for this peer review?** For information about this choice, including consent withdrawal, please see our Privacy Policy .

Reviewer #1: No

Reviewer #2: No

---

## [Author Response · Author response to Decision Letter 1]

4 Nov 2024

EDITOR’S COMMENTS

- Authors should address the reviewers comments in the attachment.

RE: Thank you. We have now addressed all the comments raised by the reviewers. Where necessary, we have made substantial revisions; however, in areas where we differ, we have provided rebuttal to the comments concerned. Also, our comments are in bold fonts. Thank you.

- Include a critical assessment of the individual sources of evidence to determine their quality and bias.

RE: We have now done a quality appraisal (risk of bias assessment) of all the included sources of evidence. Thank you.

- Authors should comment (in the discussion) on the areas for potential systematic reviews on this topic in the future.

RE: Potential areas for systematic reviews have now been added. Thank you.

REVIEWERS’ COMMENTS

Reviewer #1: The manuscript was well written, the methodology was very robust and result presentation highly scientific. Additionally, the process for reporting systematic reviews and meta-analysis was strictly followed.

RE: Thank you very much for the comment.

Reviewer #2: Thank you for the opportunity to review your work. I commend the effort. See the attachment for my comments.

RE: ORAL CANCER IN EAST AFRICA: A SCOPING REVIEW OF EMPIRICAL EVIDENCE

introduction

Line 84 Rephrase: "The rising incidence of oral cancer in the region is largely due to risk factors, including tobacco and alcohol use...."

The risk factors should be qualified.

"largely due to increased exposure to risk factors, including tobacco and alcohol use...."

RE: Thank you. We have now rephrased this text.

Many sentences also need to be written better. Line 92: Late detection of oral cancer is a major challenge in East Africa.

rephrase the sentence. late detection is a common occurrence in east Africa and a challenge

RE: Thank you. We have now adopted your recommendation.

is this a scoping review: generally, should be more streamlined. oral cancer in east Africa a scoping review of empirical evidence (key oral cancer, status, empirical data) VS construction of ML models for oral cancer outcomes. (three things here outcome, models and cancer). status is so wide. can we streamline it to outcomes. status is everything from diagnosis to risk to outcomes. importantly, to map out rapidly the key concept in an area (oral cancer status in africa), the sources and type of evidence available. because it has not been

RE: Thank you. We have now streamlined the title of our scoping review. The new title is Epidemiology, Literacy, Risk Factors, and Clinical Status of Oral Cancer in East Africa: A Scoping Review.

introduction

"The rising incidence of oral cancer in the region is largely due to risk factors, including tobacco and alcohol use...."

"largely due to increased exposure to risk factors, including tobacco and alcohol use...."

RE: Thank you. We have now rephrased this text.

92: Late detection of oral cancer is a major challenge in East Africa. rephrase the sentence. late detection is a common occurrence in east Africa and a challenge

RE: Thank you. We have now adopted your recommendation.

line 110: Map is a weak verb to use in the aim.

RE: Thank you. We have now changed the word “map” to this phrase “describe, synthesize, and appraise”.

While there are many studies available, it would be better to organize your aim further into classifying these evidences based on prevention, risk factors, treatment and outcomes. Also use charts to illustrate the different

RE: Thank you. We have now done so, and we have, as a result, rephrased the title of the review.

what variables are you comparing e.g surgery vs radiotherapy, 1 year survival and 3 year survival.

RE: Thank you. This is not applicable to our scoping review scope. Our review seeks to describe, synthesize, and appraise the available evidence on oral cancer in East Africa. Our results are based on the comprehensive review of existing findings. Thank you.

146: I could not locate "search strings used to search each of the databases" in the supplement

RE: Thank you. Please kindly see Tables S1 to S3 of the supplementary file for the search strings. They are in those tables. Thank you.

METHODS

Why did the authors not use AJOL, African journal online database

RE: Thank you. AJOL lacks a good search engine that was why we did not use it. We used NINE major databases for this review, which is very sufficient. According to AMSTAR-2 guidelines, a minimum of TWO databases is considered sufficient for a structured literature review.

1. A general write-up of what the study method is going to be like should be stated. this should included participants, comparison, outcome (PECO)

RE: We stated it in the manuscript. We stated that we used the Arksey and O’Malley’s framework for the conduct of the study. Also, it is more appropriate to use PCC framework in scoping reviews, instead of PECO framework. Please see this article as evidence to support our argument.

Khalil, H., & Tricco, A. C. (2022). Differentiating between mapping reviews and scoping reviews in the evidence synthesis ecosystem. Journal of clinical epidemiology, 149, 175–182. https://doi.org/10.1016/j.jclinepi.2022.05.012

We have now used the PCC framework instead, and we used it to define the research question of the scoping review. Thank you.

Line 130-146: can you put the search terms as a string which can be validated. This was not seen in the supplemental data

RE: Thank you. Please kindly see Tables S1 to S3 of the supplementary file for the search strings. They are in those tables. Thank you.

4. mention more about the articles used in terms of:

a. Studies with suspected data fabrication/falsification,

RE: Thank you. Since we do not have access to the raw data used in the reviewed studies, it will not be fair or appropriate to accuse the author(s) of any of the included study on data fabrication/falsification. Thank you.

4. mention more about the articles used in terms of:

b. incomplete reporting to assess, stratify oral cancers, Duplicate studies (data from the same center but different years),

RE: Thank you. We do not have any of the included articles that was a duplicate study. Concerning the stratification of oral cancers, we have done our best to provide the information based on the available data in the included articles. Thank you.

The agreement between both reviewers in selecting articles using kappa values would be good.

RE: Thank you. All screenings were done by two reviewers while a third reviewer resolved conflicts. We did not calculate the kappa values because the three reviewers involved had published extensively on oral cancer and as well scoping/systematic reviews. Thank you.

5. Data collation: the author names, publication year, study country, study design, sample size, cancer types, outcomes should be included in the table.

RE: Thank you. All these information except sample size and cancer types have already been stated in Table 1. We have now included them in Table 1. Thank you.

6. Risk of Bias: though not compulsory, it is good for a high level journal.

RE: We have now done a quality appraisal (risk of bias assessment) of all the included sources of evidence. Thank you.

7. statistical analysis protocol was not mentioned

what did you do about missing data, Descriptive statistics calculated as median, interquartile range, and frequencies, use of pearson’s Chi-square test etc. Probability values and statistical tool used.

RE: Thank you. Our study was a scoping review; it was neither a meta-analysis nor a cross-sectional analytical study. Hence, the inclusion of such information is out of scope/place in our study. Thank you.

RESULTS

A. General considerations

Figure 1 : Reasons for excluding some text in the flow chart. Any duplication in the full text?

list citations in a table as supplemental data. flow cart why was 77 papers removed. why was a report not retrieved what do you mean by old publication considering you did a search based on time/date limit

RE: Thank you. Those 77 papers were excluded at title and abstract screening stage, and it is a universally known fact reasons for exclusion at the title and abstract screening stage are not needed in the PRISMA flow chart. The only stage where such reasons need to be stated are at the full text screening stage which we did. Please kindly see a shot/snap of the box where the reasons were stated below. Thank you.

B. General characteristics of studies included: A single table did not do justice to the data you extracted from the articles. Perhaps you can improve the work by

1. figure for years of publications

RE: Figure for years of publications has been added

2. figures of countries involved in the study. a country chart with the frequency of studies

RE: Figure of countries included in the review has now been added.

3. statement about where the studies were mostly from.

RE: Statement about where the studies were mostly from can be found on lines 246-259

4. study design. use a chart

RE: The study design is now presented using a chart

5. which studies reported the study objectives. how many reported risk/aetiology in oral cancer, outcomes of oral cancer etc

i. commonest objectives reported e.g risks, treatment.

the three commonest themes were risk(30%), treatment(25%)

The tables of all the studies should be broken down into charts to illustrate key finding from the study. e.g charts for frequency of study by years, charts for study design, maps to show countries involved and commonest anatomic sites and lesions. make it more reader friendly.

RE: These have been addressed as mentioned above, in addition, a chart showing the distribution of included articles by type of cancer investigated and research objectives were also added.

I found it difficult going through the list and am sure many readers will too.

any statistical relationship between the variables?

RE: Analysis of statistical relationship between variables is beyond the scope of this study.

C. evaluating the themes: I do not think the themes were properly developed. The themes should be discussed along well known themes in oral cancer such as risk factors, treatment, prognosis, pathogenesis etc.

RE:

Risk of bias assessment for studies would have been good

while this is optional, I believe it should be included in this work, considering the evidences available are diverse and heterogeneous. Also to get published in a journal of high impact, it will be wise to include all optional requirements.

RE: We have now done a quality appraisal (risk of bias assessment) of all the included sources of evidence. Thank you.

DISCUSSION

1. A brief summary of the purpose of this review would have been appropriate as the first few paragraphs

e.g Appraising the existing evidence of oral cancer in east Africa is crucial to improving the prevention, diagnosis, and treatment of cancer considering a predicted rise in disease incidence and mortality

RE: A brief summary of the overarching purpose of the manuscript has now been inserted as an introductory statement. Thank you.

2. a general statement about the status of the evidence is required with comparison with global trends. with further commentary on the majors themes and countries.

RE: Such a statement has now been inserted. Thank you.

This review found a recent increase in the publication of oral cases which is in line with global trends. However most of these reports were from Sudan (this is important because it shows where the data is coming from). it’s also important to discuss the result by giving reasons for the discrepancies.

RE: This has now been addressed in the text. Thank you.

3. There is a lot of information from the review. It would be better to have them organized in a manner that can be easily followed and referenced.

As mentioned earlier, it is better to arrange your discussion along certain themes that can be easily understood and followed. Risks, screening, aetiology, prevention, outcomes, prognosis, treatment. etc.

RE: The discussion has now been arranged to follow the main thematic areas identified in our result section. Thank you.

The microbiome might be discussed under risks or pathogenesis. The reason for discussing these finds from your search should be stated.

Genetic/molecular works should be clear. This gives a view on what has been done in that region rather than lumping all the information together.

RE: Microbiome-related findings have now been pooled into the segment discussing the risks for oral cancer as reported in reviewed publications. Thank you.

What is the role of OKC in this manuscript. OKC is not a cancer

Effort should be made to compare the malignancies in these articles

RE: Thank you. We have now removed the information on OKC, and we have revise the text, where applicable, to compare malignancies.

your definition of oral cancer in the introduction is quite different from those in the discussion where you seem to have lumped up oral and Maxillofacial cancer, head and neck cancer, including nasopharyngeal cancers. Kindly clarify

RE: This is a general problem encountered in relation to oral cancer. Studies are heterogeneous in nature when it comes to the definition of oral cancer and authors use terms like oral cancer, oral cavity cancer, oropharyngeal cancer, mouth cancer and also OSCC. What we have given in the introduction is the definition that is fit for the review but what we have described in the results and discussion is what other authors have used. We included all these terms in our inclusion criteria for the search to capture as many papers as possible. Thank you.

Line 568: “Some of the factors found to be responsible include gender, occupation, smoking behaviour, self-detection of symptoms, personal beliefs about symptoms, interpretation of diagnosis, limited access to health”

Kindly qualify gender, occupation. e.g male gender, occupation? how and which one?

RE: This information has now been added. Thank you.

Add limitations of the study

RE: Limitations of the study can be found in the discussion section, highlighted. Thank you.

---

## [Decision Letter · Decision Letter 1]

23 Dec 2024

Epidemiology, Literacy, Risk Factors, and Clinical Status of Oral Cancer in East Africa: A Scoping Review

PONE-D-24-22531R1

Dear Dr. Murererehe,

We’re pleased to inform you that your manuscript has been judged scientifically suitable for publication and will be formally accepted for publication once it meets all outstanding technical requirements.

Kind regards,

John Adeoye

Academic Editor

PLOS ONE

Additional Editor Comments (optional):

Reviewers' comments:

Reviewer's Responses to Questions

**Comments to the Author**

1. If the authors have adequately addressed your comments raised in a previous round of review and you feel that this manuscript is now acceptable for publication, you may indicate that here to bypass the “Comments to the Author” section, enter your conflict of interest statement in the “Confidential to Editor” section, and submit your "Accept" recommendation.

Reviewer #2: All comments have been addressed

2. Is the manuscript technically sound, and do the data support the conclusions?

Reviewer #2: Yes

3. Has the statistical analysis been performed appropriately and rigorously? 

Reviewer #2: Yes

4. Have the authors made all data underlying the findings in their manuscript fully available?

Reviewer #2: Yes

5. Is the manuscript presented in an intelligible fashion and written in standard English?

Reviewer #2: Yes

6. Review Comments to the Author

Reviewer #2: Thank you for responding to my comments. A lot of improvement has been made, it's easier to follow and read.

7. PLOS authors have the option to publish the peer review history of their article (what does this mean? ). If published, this will include your full peer review and any attached files.

**Do you want your identity to be public for this peer review?** For information about this choice, including consent withdrawal, please see our Privacy Policy .

Reviewer #2: No

---

## [Editor Report · Acceptance letter]

PONE-D-24-22531R1

PLOS ONE

Dear Dr. Murererehe,

I'm pleased to inform you that your manuscript has been deemed suitable for publication in PLOS ONE. Congratulations! Your manuscript is now being handed over to our production team.

Kind regards,

on behalf of

Dr. John Adeoye

Academic Editor

PLOS ONE